# Investigation of the Familial Risk of Rheumatic Heart Disease with Systematic Echocardiographic Screening: Data from the PROVAR+ Family Study [note 1]

**DOI:** 10.3390/pathogens11020139

**Published:** 2022-01-24

**Authors:** Juliane Franco, Bruno R. Nascimento, Andrea Z. Beaton, Kaciane K. B. Oliveira, Marcia M. Barbosa, Sanny Cristina C. Faria, Nayana F. Arantes, Luana A. Mello, Maria Cecília L. Nassif, Guilherme C. Oliveira, Breno C. Spolaor, Carolina F. Campos, Victor R. H Silva, Marcelo Augusto A. Nogueira, Antonio L. Ribeiro, Craig A. Sable, Maria Carmo P. Nunes

**Affiliations:** 1Serviço de Cardiologia e Cirurgia Cardiovascular e Centro de Telessaúde do Hospital das Clínicas da UFMG, Belo Horizonte 30130-100, MG, Brazil; francojulianef@gmail.com (J.F.); kacianebruno@hotmail.com (K.K.B.O.); marciamelobarbosa@gmail.com (M.M.B.); sannyfaria@gmail.com (S.C.C.F.); nayanaflamini@yahoo.com.br (N.F.A.); luanaguiarmm@gmail.com (L.A.M.); cecilialn2@gmail.com (M.C.L.N.); guilhermecatizani@gmail.com (G.C.O.); breno.spolaor@hotmail.com (B.C.S.); carolinafcampos@outlook.com (C.F.C.); victorraggazzi@hotmail.com (V.R.H.S.); mar.augusto@yahoo.com (M.A.A.N.); tom1963br@yahoo.com.br (A.L.R.); mcarmo@waymail.com.br (M.C.P.N.); 2Departmento de Clínica Médica, Faculdade de Medicina da Universidade Federal de Minas Gerais, Belo Horizonte 30130-100, MG, Brazil; 3The Heart Institute, Cincinnati Children’s Hospital Medical Center, School of Medicine, University of Cincinnati, Cincinnati, OH 45229, USA; Andrea.Beaton@cchmc.org; 4Cardiology, Children’s National Health System, Washington, DC 20010, USA; CSABLE@childrensnational.org

**Keywords:** rheumatic heart disease, echocardiography, screening, family, risk

## Abstract

We aimed to use echocardiographic (echo) screening to evaluate the risk of Rheumatic Heart Disease (RHD) among the relatives of patients with advanced RHD, who were enrolled in the University Hospital’s outpatient clinics from February 2020 to September 2021. Consenting first-degree relatives were invited for echo screening using handheld devices (GE VSCAN) by non-physicians, with remote interpretation. Matched controls (spouses, neighbors) living in the same household were enrolled in a 1:5 fashion. A standard echo (GE Vivid-IQ) was scheduled if abnormalities were observed. In 16 months, 226 relatives and 47 controls of 121 patients were screened, including 129 children, 77 siblings and 20 parents. The mean age was 40 ± 17 years, 67% of the patients were women, and 239 (88%) lived with the index case for >10 years. Echo findings suggestive of RHD were confirmed in zero controls and 14 (7.5%) relatives (*p* = 0.05): 11 patients had mild/moderate mitral regurgitation, and four were associated with mitral stenosis and abnormal morphology. Two patients had mild aortic regurgitation and abnormal morphology, which were associated with mild aortic and mitral stenosis, and two patients with advanced RHD had bioprostheses in the mitral (2) and aortic (1) positions. In conclusion, first-degree relatives of individuals with clinical RHD are at greater risk of having RHD, on top of socioeconomic conditions.

## 1. Introduction

The global burden of Rheumatic Heart Disease (RHD) is still high, and among cardiovascular diseases, it accounts for 1.6% of all deaths, resulting in 306,000 deaths yearly worldwide [1]. The epidemiological improvement of RHD over the past decades was unevenly distributed, with a considerable reduction of prevalence and mortality being observed in high-income regions, contrasting with a stable or worsening pattern in low-income endemic countries, where 80% of the children at-risk live.

In Brazil, valvular dysfunction from RHD is responsible for nearly 50% of valve surgeries in the public health system [2]. It is noticeable, however, that the age-standardized prevalence of RHD showed a stable pattern from 1990 to 2019, and a remarkable 59% reduction of age-standardized mortality was observed [1]. This resulted not only from socioeconomic development, but also reflects the expansion of the public health system and improved access to basic RHD care (e.g., the treatment of pharyngitis and acute rheumatic fever, primary and secondary prophylaxis, and clinical follow-up), provided at the primary level. Public secondary and tertiary care—including surgery—are also available, although they are unequally distributed in the territory [3]. Despite such advances, recent school-based screening studies showed a high burden of subclinical RHD (around 4.5%) that parallels countries with worse indexes of socioeconomic development [4,5]. Thus, the optimization of active case finding strategies is of the utmost importance, in order to better define individuals who are at high risk, and to guide diagnostic approaches.

RHD results from a complex interaction between the socioeconomic environment and host susceptibility. Relatives of patients with advanced RHD share both, and may be at high risk. The genetic predisposition for the development of RHD remains incompletely understood, but new data emerging from genome-wide association studies have identified several immune-related polymorphisms, including in Human Leukocyte Antigen and immunoglobulin heavy chain loci, that may modify the immune response [6,7,8].

There is some evidence to suggest that screening relatives of patients identified with RHD may be a high-yield active case detection strategy. In Uganda, a targeted echo screening study demonstrated that screen-positive siblings of RHD-positive cases are more likely to have definite RHD, which was noticeable if the index case fulfilled the “definite” criteria [9]. In this study, we aimed to use echocardiographic (echo) family screening to evaluate the risk of RHD among first-degree relatives of patients with advanced clinical RHD, as compared to non-relatives sharing a similar household or family compound.

## 2. Materials and Methods

The PROVAR+ study is a continuation of the rheumatic heart disease (RHD) program established in 2014 as a collaboration between the Universidade Federal de Minas Gerais (UFMG), the Telehealth Network of Minas Gerais [10] and the Children’s National Health System, Washington, DC, USA. The study’s methodology and results have been detailed elsewhere [5]. In summary, the study’s methodology is based on the utilization of non-experts for the acquisition of echo screening images, on handheld (VScan^®^, GE Healthcare, Milwaukee, WI, USA) devices, and telemedicine remote image interpretation by experts in Brazil and the US, applying the 2012 World Heart Federation (WHF) criteria [11] for RHD. The screening was initially focused on schoolchildren from schools located in underserved areas of Southeast Brazil, and the program was then expanded to primary care, aimed at the early diagnosis of heart disease in adults and the elderly, as well as the prioritization of care in resource-limited settings [4,5,12,13].

In this sub-study, from February 2020 to September 2021, patients with known advanced RHD, with valvular involvement confirmed by clinical examination and echocardiography, were consecutively enrolled in the outpatient clinics of UFMG University Hospital—a quaternary public institution with 509 beds and 1500 admissions/month, located in Belo Horizonte, MG, Southeast Brazil. First-degree relatives were consented and invited for echo screening by non-physicians (the 2012 WHF Criteria [11] and the American Society of Echocardiography (ASE) criteria adapted for the absence of spectral Doppler [14,15,16]) using handheld devices (GE VSCAN, Horten, Norway), with telemedicine consensus interpretation by 2 experts (board-certified cardiologists from the University’s staff, trained in previous phases of the study), and a tie-breaker in case of discrepancies. Socioeconomically-matched controls (spouses, neighbors in the same family compound) living in the same household for at least 5 years, and thus sharing a similar socioeconomic environment, were also enrolled in a 1:5 fashion (Figure 1). The screening team consisted of one nurse and one physical therapist, who were previously trained with a combination of online RHD educational modules (http://www.wiredhealthresources.net/EchoProject/index.htm; accessed on 5 May 2021) and at least 6 weeks of hands-on training.

Prior to the screening, a detailed protocol including demographics and self-reported socioeconomic and clinical variables (comorbidities, current and past medications, history of cardiovascular disease, past diagnosis of ARF, RHD and/or secondary prophylaxis) was applied by the study team. Ethics approval was obtained from the Universidade Federal de Minas Gerais Institutional Review Board and the Belo Horizonte City Board of Health, and all of the patients included in this analysis signed an informed written consent form prior to enrollment, in the first in-person consultation.

For the screening, the participants underwent a simplified 7-view protocol—performed in the outpatient clinics during regular appointments, using examination stretchers—focused on mitral, aortic and tricuspid valves, left and right ventricular morphology and function, and pericardial effusion. Objective and subjective observations were reported. A locally developed cloud system (SigTel^®^, Universidade Federal de Minas Gerais, Belo Horizonte, MG, Brazil), and GE proprietary offline software (Gateway^®^) were used for the telemedicine. A confirmatory standard echo with fully functional portable machines (GE Vivid IQ, Milwaukee, WI, USA), provided by experts, was scheduled if abnormalities were observed, and all of the participants found to have RHD or other structural heart disease were referred for follow-up. If they were indicated, the participants were enrolled in the University Hospital Cardiology clinics for specialized care, and the continuation of care was left to the discretion of the attending physician.

## 3. Statistical Analysis

All of the data were systematically entered into the RedCap^®^ online database [17]. The statistical analysis was performed using SPSS^®^ software version 23.0 for Mac OSX (SPSS Inc., Chicago, IL, USA). The categorical variables, expressed as numbers and percentages, were compared between the groups (first-degree relatives of individuals with clinical RHD and non-relatives) using Fisher’s exact test, whereas the continuous data, expressed as the mean ± SD or median and Q1/Q3 (25%/75%), were compared using Student’s unpaired t-test or the Mann–Whitney U test, as appropriate. Demographic, clinical and echocardiographic variables were compared between the groups, and proportions were presented with 95% confidence intervals (CI). The *k**appa* coefficient was used to assess the inter-reviewer reliability for the interpretation of the screening echoes. A two-tailed significance level of 0.05 was considered statistically significant.

## 4. Results

Over 18 months, 226 relatives and 47 controls of 121 patients were screened, including 129 children, 77 siblings and 20 parents (Figure 1). Detailed characteristics of the study groups are provided in Table 1. The mean age was 40 ± 17 years, 67 (61.2%) of the participants were women, and 239 (87.5%) had lived with the index case for >10 years. Among the first-degree relatives, 26 (11.5%) had more than one family member with a history of RHD. Among the index cases, 70.8% reported the past prescription of Benzathine Penicillin B (BPG), but only 21.1% were currently under prescription. The clinical characteristics and previous medical history were similar between the groups, except for hypertension and female sex being more frequent among the relatives (18.6% vs. 31.9%, *p* = 0.049 and 67.7% vs. 29.8%, *p* < 0.001), along with the notable older age of the control group (Table 1). The maternal literacy of the screened relatives was overall low, with a high proportion of illiterates and/or with incomplete elementary school.

An abnormal screening echocardiogram prompted complete echocardiographic assessment in 43 participants (15.9% overall; 17.4% relatives vs. 8.7% controls, *p* = 0.18). Screening echo findings preliminarily suggestive of RHD were observed in zero controls and 17 (7.5%, 95% CI 4.4–11.8) relatives (*p* = 0.05). Among these patients, all had mitral valve (MV) involvement: 14 (82.4%) had mild-to-moderate mitral valve (MV) regurgitation, and six had signs of MV stenosis. Four patients (23.5%) had mixed valve involvement in the screening, with mild-to-moderate regurgitation of the aortic valve (AV), with additional findings suggestive of AV stenosis in two cases. The overall *kappa* between the first and second echo reviews for the presence of RHD was 0.89. Detailed echocardiographic findings are provided in Table 2.

The presence of RHD findings was confirmed in the standard echo in 14 patients (6.2%, 95% CI 3.4–10.2) (82% agreement with the screening findings). Eleven had MV disease and three had mixed (mitral and aortic valve) disease. All 11 patients had mild-to-moderate MV regurgitation, four with associated MV stenosis and abnormal morphology (including two bioprostheses). Two patients had mild AV regurgitation and abnormal AV morphology (leaflet thickening) associated with mild AV and MV stenosis, and two patients with a previous history of advanced RHD had bioprostheses in the MV (2) and AV (1) without significant dysfunction at the time of screening (Table 2).

Notably, one case of mild left ventricular dysfunction was observed. Another patient had an indication of commissurotomy for MV stenosis at the time of diagnosis (Figure 2). The indication of commissurotomy for this patient occurred at the same time as that of the index RHD case (sister), and the procedure was successfully performed in both, with full recovery. The remaining RHD patients were enrolled in specialized clinical follow-up.

## 5. Discussion

Our data, which were derived from systematic echo screening in a specialized tertiary care center, suggest that first-degree relatives of patients with advanced RHD have a high risk of having RHD, even compared to non-relatives sharing a similar household and socioeconomic environment. The RHD phenotypes at diagnosis varied from predominantly subclinical valvular disease, to advanced sequelae with the indication of interventions. The findings, coupled with previously published studies [9,18], point towards the inclusion of first-degree relatives as priorities in targeted screening programs.

As systemic rheumatic diseases affect the heart heterogeneously, depending on the disease stage and several host factors, multimodality imaging is proposed for the diagnosis and risk stratification of these patients [19]. In this scenario, echocardiographic screening for RHD has been explored as a tool for early case detection and epidemiological surveillance for more than a decade. Broad screening programs have been limited, as non-targeted approaches are challenging to scale and have questionable cost-effectiveness [20,21,22,23]. Furthermore, even with the growing utilization of cost-saving approaches such as task-shifting and telemedicine for remote interpretation [24], the availability of personnel and resources remains an issue in the least-resourced settings, urging more selective strategies for the improvement of early diagnosis.

In this scenario, there has been much research interest in the familial/genetic predisposition to RHD—especially to the most severe phenotypes. In Brazil, studies about the association between genetic polymorphisms and cytokine expression, and unfavorable outcomes of latent and clinical RHD (progression to clinical disease and severe valve involvement requiring intervention, respectively) have been conducted. Whilst, among individuals with latent RHD, interleukins (IL-4, IL-8 and IL-1RA) seem to predict clinical disease, in patients with established RHD the co-regulated expression of IL-6 and TNF-α is associated with severe valve dysfunction, and high IL-10 and IL-4 levels predict adverse outcomes [25]. When samples from individuals with latent and clinical RHD and matched controls were compared, higher levels of all of the cytokines associated with clinical compared to latent RHD—these being IL-4, CXCL8 and IL-1RA—were the strongest predictors of clinical disease. Additionally, polymorphisms in the IL-2, IL-4, IL-6 and IL-10 genes associated with clinical RHD, and the discriminative value of IL-4 (both gene polymorphism and phenotypic expression), to differentiate between latent and clinical RHD were reinforced [26]. Although they were not conclusive, these local data add to the body of evidence that supports genetic predisposition to RHD, which is aligned with the findings of our family study.

Clinical studies also support family predisposition to RHD on top of a shared socioeconomic environment. A meta-analysis including 435 twin pairs—mostly from North America and Ireland—between 1933 and 1964 showed a pooled concordance risk for acute rheumatic fever (ARF) of 44% in monozygotic twins and 12% in dizygotic twins (OR 6.39), with an estimated heritable risk of 60% [27]. The family concordance for RHD among first degree relatives of 70 index cases of ARF was also demonstrated in New Zealand: 94 parents and 132 siblings of 70 index cases were screened, and the RHD prevalence (42/1000 and 90/1000, respectively) was much higher compared to the high ARF incidence populations in the country [18].

Contrasting with this sample of individuals with advanced disease and clinical manifestations, 60 patients with echocardiography-detected RHD (borderline and definite) were enrolled in targeted family screening in Uganda; compared to the controls, definite RHD was 4.5 times more common in the siblings of RHD-positive individuals, reaching 5.6 times when only index cases with definite RHD were considered [9].

Our study adds to this field, including index cases with the more severe spectrum of clinical RHD, which are presumably a group with a higher predisposition to continuing valvular damage and inflammation. The competing role of a shared household and socioeconomic status was controlled by the inclusion of non-relatives sharing the same environment for a considerable time period (at least 5 years). Additionally, the imaging flowchart merged the practical task-shifted screening with ultraportable devices, with confirmation from fully functional devices operated by experts, with good agreement for the RHD findings (confirmed in 14 out of 17 screen-positive individuals). Even considering the milder RHD phenotypes found in screening studies in Brazil—as compared to other endemic regions [28]—our results showed, in addition to a surprisingly high prevalence, severe cases among family members. Out of 14 relatives with confirmed RHD, one had prompt indication to intervention, one was included in close monitoring, and two had prosthetic valves.

There are particular characteristics of this sample that may have influenced our findings. The index cases were selected from a population with access to a public tertiary specialized outpatient clinic, which may reflect a socioeconomically privileged strata of the Brazilian population. However, the median household in both groups (5.5 ± 2.4 vs. 4.9 ± 3.9/house)—a surrogate for socioeconomic status and overcrowding—was considerably higher than the Brazilian average (2.9/house) even if it was compared to the highest rates of the northern region (3.3/house) [29], reinforcing the vulnerability of our population. In terms of clinical profile, hypertension was more prevalent in the control group, which was possibly associated with its older age. Despite the presumable impact of risk factors on progression and the severity of valve heart disease [1], the prevalence of hypertension in our population was close to that observed in a meta-analysis (28.7%), in the National Health Survey (32.3%), and in the more conservative GBD estimates for adults (18.9%) [30]. The higher rates of previous stroke among the relatives, on the other hand, was probably a result of valvular involvement.

Since the implementation of the first large-scale RHD screening programs in Brazil, in 2014, consent to participate and adherence have been major challenges. Despite the provision of educational curricula in schools prior to any research interventions, the rates of informed consents signed by parents was universally low (<40%) [5]. However, when the family interactions were carried out by health agents in the existing primary care program, the numbers improved considerably (to over 80%) [5]. Similarly, a high degree of family acceptability of school-based echo screening was observed in a high-risk population in New Zealand [31], and these findings were confirmed during targeted ambulatory family screening [18], suggesting a potential for more targeted and personalized interventions. Furthermore, screening for different purposes in the Brazilian primary care, supported by the Family Health Program—a successful community-based approach [3]—has proven to be feasible, with high participation and interaction with the covered population [12,13]. This, in addition to the high prevalence observed, points towards the possibility of expanding our strategy—initially deployed in a tertiary university hospital—to primary care, improving case-finding and providing access to the families at highest risk. However, additional studies are warranted, with larger samples, including a wider variety of presentations—such as latent and post-surgical RHD—along with the longitudinal evaluation of the impact of the family intervention. In addition, cost-effectiveness analyses in these different scenarios are also essential to guide discussions about the implementation of the strategy.

## 6. Limitations

Our study has several limitations, most of them inherent to the difficulties in enrolling a large sample of RHD patients which are representative of the Brazilian population in terms of age, gender and race distribution. First, our sample size, which was included by convenience in the 18-month study period, was limited to 226 relatives and 47 controls, precluding more robust analyses such as the multivariable assessment of RHD predictors. Although limited, this sample consists of a high-risk population selected in a tertiary specialized outpatient clinic—a referral center for the state—allowing for novel insights about familial predisposition among the individuals with the most severe phenotypes. Second, also due to sample limitations, no stratified sampling procedures were carried out; thus, the findings cannot be extrapolated to the Brazilian population. The purpose of the control group was to include patients based on similar (or equal) socioeconomic backgroundsand households—key drivers of RHD prevalence—and, thus, no pairing was possible for sex and age. Third, the baseline screening was performed by non-physicians utilizing ultraportable handheld devices. Although this may limit accuracy, considering personnel training and the absence of advanced spectral Doppler capabilities, task shifting, and the use of low-cost equipment are recognized strategies for the expansion of screening programs in under-resourced settings, which was the rationale of this study. Furthermore, the method tends to be more sensitive than specific, and the indication of a fully functional standard echo for confirmation was made even in the presence of minor findings. Fourth, the lower proportion of women in the control group may have underestimated the prevalence, considering the more frequent incidence of RHD among females. Lastly, no inferences about cost-effectiveness can be drawn from our preliminary data. Even with the aforementioned limitations, to the best of our knowledge this is the first screening program in South America targeted to family members and individuals sharing a similar socioeconomic environment; it allows for inferences about the familial risk of RHD, in addition to available data from other settings.

## 7. Conclusions:

First-degree relatives of individuals with clinical RHD are at greater risk of having RHD—even if compared to non-relatives sharing the same social environment—and family screening should be considered in high-risk and endemic populations. Genotyping studies are warranted to better understand individual and family susceptibility to RHD, in addition to socioeconomic conditions and other known drivers of disease.

## Figures and Tables

**Figure 1 pathogens-11-00139-f001:**
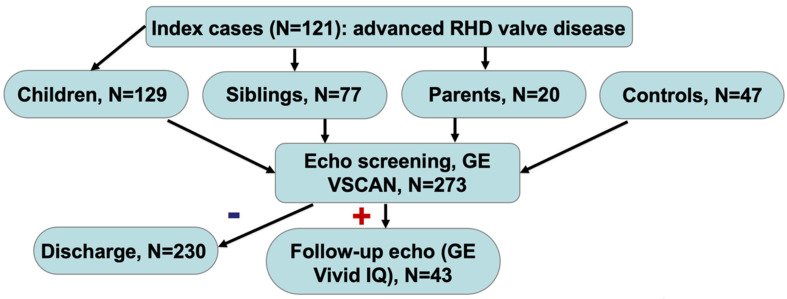
Operational flowchart of the PROVAR+ family study. Abbreviations: GE, general electric; RHD, rheumatic heart disease.

**Figure 2 pathogens-11-00139-f002:**
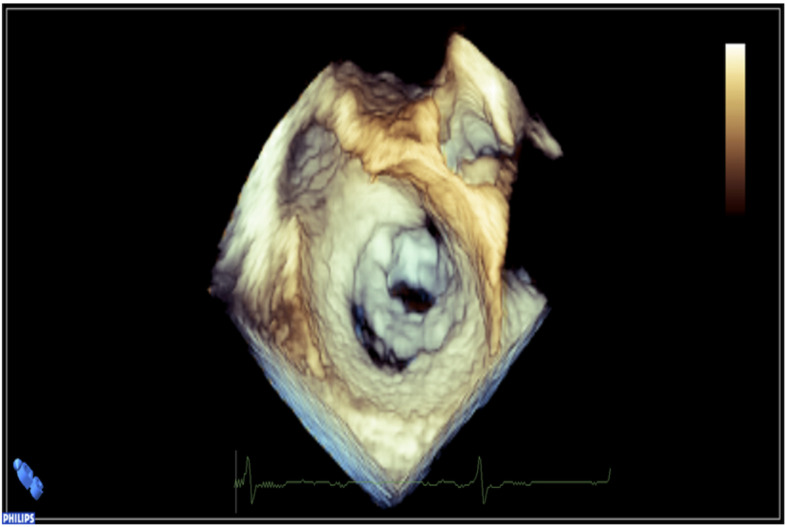
A 3D confirmatory standard echocardiogram showing severe mitral valve stenosis, associated with marked morphological abnormalities in a first-degree relative with the indication of mitral commissurotomy at the time of diagnosis. The index case (sister) had a similar valvular condition with a concomitant indication of intervention.

**Table 1 pathogens-11-00139-t001:** Demographic and clinical characteristics of the first-degree relatives and controls included in the echocardiographic screening.

Variables:	Relatives (N = 226)	Controls (N = 47)	*p*-Value
Age (years, mean ± SD)	37.8 ± 16.6	52.2 ± 16.3	**<0.001***
Sex (female, N (%))	153 (67.7)	14 (29.8)	**<0.001***
Household (mean ± SD)	5.5 ± 2.4	4.8 ± 3.9	0.23
Origin (rural/small town) (N, %)	106 (46.9)	28 (59.6)	0.28
Mother’s education (illiterate/incomplete elementary school) (N, %) †	166 (73.5)	42 (89.4)	0.08
Kinship (N, %):			
• Siblings	• 77 (34.1)		N/A
• Children	• 129 (57.1)		
• Parents	• 20 (8.8)		
• Other (controls)		• 47 (100)	
Living with index case (N, %):			
• Up to 10 years	• 21 (9.3)	• 10 (21.3)	**0.013 ***
• 10 to 20 years	• 85 (37.6)	• 8 (17)	
• Over 20 years	• 118 (52.2)	• 28 (59.6)	
Hypertension (N, %)	42 (18.6)	15 (31.9)	**0.049 ***
Diabetes (N, %)	18 (8.0)	1 (2.1)	0.21
Known history of RHD (N, %)	6 (2.7)	0	0.59
Stroke (N, %)	4 (1.8)	1 (2.1)	1.00
Previous symptoms of heart failure (N, %)	4 (1.8)	0	1.00
Known history of coronary artery disease (N, %)	5 (2.2)	3 (6.4)	0.14
Recurrent pharyngitis (N, %) ‡	65 (28.8)	8 (17.0)	0.08
**Symptoms and clinical presentation:**			
Dyspnea (N, %)	76 (33.6)	7 (14.9)	**0.014 ***
Chest pain (N, %)	72 (31.9)	9 (19.1)	0.11
Palpitations (N, %)	73 (32.3)	11 (23.4)	0.30

Abbreviations: BPG: Benzathine Penicillin G; RHD: rheumatic heart disease. * *p* < 0.05. † The proportion who were reportedly illiterate or with incomplete elementary school; ‡ the reported occurrence of ≥2 episodes of pharyngitis in a 1-year period.

**Table 2 pathogens-11-00139-t002:** Echocardiographic characteristics of the first-degree relatives and controls included in the echocardiographic screening.

Variables:	Relatives (N = 226)	Controls (N = 47)	*p*-Value
**Screening echocardiography (N = 273)**			
LV dysfunction (mild) (N, %)	1 (0.4)	0	1.00
LV hypertrophy (mild/moderate) (N, %)	15 (6.6)	7 (14.9)	0.08
Mitral valve (N, %):			
• Rheumatic mitral valve	• 15 (6.6)	• 0	0.17
• Mitral valve prolapse	• 3 (1.3)	• 0	
• Other	• 4 (1.8)	• 2 (4.3)	
Mitral regurgitation (mild/moderate) (N, %)	52 (23.0)	9 (19.1)	0.70
Mitral stenosis (N, %)	6 (2.7)	0	0.59
Aortic valve (N, %):			
• Rheumatic aortic valve	• 3 (1.3)	• 0	0.54
• Calcific aortic valve	• 9 (4.0)	• 2 (4.3)	
• Other	• 7 (3.1)	• 0	
Aortic regurgitation (mild/moderate) (N, %)	26 (11.6)	5 (10.6)	1.00
Aortic stenosis (N, %)	7 (3.1)	0	0.61
Tricuspid regurgitation (N, %)	37 (16.5)	6 (12.8)	0.66
Indication for standard echo (N, %)	39 (17.4)	4 (8.7)	0.19
RHD (suggestive) (N, %)	17 (7.5)	0	0.05
**Standard echocardiography (confirmed RHD cases, N = 14)**			
Valve involvement (N, %):			
• Mitral valve (isolated)	• 11 (4.9)	-	N/A
• Aortic valve (isolated)	• 0	-	
• Mixed (mitral + aortic)	• 3 (1.3)	-	
MV disease (N, %):			
• MR (mild/moderate)	• 11 (4.9)	-	N/A
• Morphological/prosthesis + MS	• 4 ((1.8)	-	
AV disease (N, %):			
• AR + AS + morphological	• 2 (0.9)	-	N/A
• Morphological (prosthesis)	• 1 (0.4)	-	

Abbreviations: AR: aortic regurgitation; AS: aortic stenosis; MR: mitral regurgitation; MS: mitral stenosis; RHD: rheumatic heart disease.

## Data Availability

The data analysis methods and study materials will be made available to other researchers for the purposes of reproducing the results or replicating the procedure, from the corresponding author upon reasonable request.

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
