# Peer review of "Investigation of the Familial Risk of Rheumatic Heart Disease with Systematic Echocardiographic Screening: Data from the PROVAR+ Family Studyâ€"

_pathogens, 2022, doi:10.3390/pathogens11020139_

Round 1
Reviewer 1 Report
Evaluation Of Familial Risk of Rheumatic Heart Disease With 2 Systematic Echocardiographic Screening: Data From the 3 PROVAR+ Family Study
Major comments
This is largely a well written manuscript which demonstrates a need for low-cost familial echo screening among first degree relatives of RHD cases in Brazil. The methodology and conclusions appear appropriate.
The Introduction and Methods are not sufficiently detailed. In places the presentation of Results in convoluted.
In places the wording does not flow well. I have noted a few examples where this occurs below. The manuscript needs a good copy-edit from a naturalised English speaker.
Minor comments
There is a typo in the last author’s degree.
Abstract
RHD should be spelled in full at first use in the Abstract (and again in the Introduction).
If there are 47 controls and 121 patients then matched controls were not enrolled in a 1:5 fashion. Please correct throughout manuscript.
Introduction
Poor English – what is meant by ‘improvement of RHD epidemiology’?
Likewise ‘in lower bounds of the socioeconomic development’ is not worded correctly.
More information about the epidemiology of rheumatic fever and RHD and case management in the area the study is set in is needed.
Introduce the University Hospital. Where is it located? What population does it treat? How many admissions/year? Is it a tertiary hospital?
Methods
What us meant by ‘Data analytic methods’? Statistical analyses should be described in the manuscript. The first sentence is odd. Is there a Data Availability section where it could appear instead?
State The PROVAR+ study aims and consider introducing it in the Introduction. Is this the PROVAR+ study or a sub-study?
‘study flowchart’ line 78 is an inappropriate description.
Who were the non-physicians performing screening? How were they selected and trained?
What sort of patient consent was obtained? Eg. Informed written consent following an in-person consultation (and with whom)? I note this is specified below in line 109, suggest deleting above consent section and expanding it here.
Who were the ‘experts’? How were they selected and trained?
How was socioeconomic matching performed? What factors were matched on? How were these measured?
Figure 1 should state the no. discharged
Where was echo screening performed? In participants’ homes? How? Eg. Patients were asked to lie on their bed while screening took place?
Line 106 – capitalise University Hospital.
Line 125 ‘at total’ suggest replacing with ‘a total of..’
The prevalence of hypertension, diabetes and previous stroke in your study sample is really high. Mention in the Discussion whether this is normal in the regional population. The way you have presented this, it looks like you have grouped the cases in with these prevalence results as well. Please clarify who these results pertain to.
‘Among 1st 127 degree relatives, 26 (11.5%) had more than 1 family member with history of RHD and, 128 among index cases, 70.8% reported past prescription of Benzathine Penicillin B (BPG), and 129 only 21.1% are currently under prescription.’ This sentence is convoluted, break it into 2 sentences addressing different things.
Be constant with abbreviations – state it in full once in the abstract and main text, then abbreviate consistently eg. mitral valve.
Table 1
Age – specify in years
What does ‘household ‘ refer to? No. people dwelling in home? If you had the number of bedrooms, this information could be combined into a more meaningful assessment of household crowning. Consider using the Canadian National Occupancy Index if data is available.
Mother education – More information is required to interpret what this line is referring to.
How was ‘recurrent pharyngitis’ defined? (State in Methods)
Was household income assessed? How did it compare with the Brazilian median income?
Table 2.
Suggest using ‘Echocardiographic findings’ in title instead of ‘characteristics’
I don’t think the total column adds much. Suggest deleting it.
I suggest avoiding using abbreviations in the tables unless it’s really necessary
Lines 152-158 – there is no need to repeat what is in the table in the text.
In the text, spell numbers under 10 in full, Roman numerals can be used for no.s with decimal places and >10. (Line 159)
Line 159 – was this the same patient with mild left ventricular dysfunction and indication of commissurotomy for MV stenosis? Specify. Break into 2 sentences if different patients.
Figure 2 - commas in the caption are unnecessary.
Discussion
Line 172 – rewrite, ‘in its majority,’ this does not make sense.
Line 174 reference all these studies
Reference 20 is formatted oddly and you seem to be missing reference 19.
Reference all of the studies you have referred to.
Line 193 – re-write. What is meant by ‘its strongest predictors’?
Reference 22 – in what population?
‘Contrasting with this sample of clinically diagnosed disease..’ rephrase.
Line 216 – ‘…inclusion of non-relatives sharing the same environment for a considerable time period…’ please define the min. time in shared environment for controls to be eligible in the Methods. Do you mean shared household not environment?
Line 245 – at first? What changed? Rewrite?
Your cases were selected from people who have access to a tertiary specialized outpatient clinic. Does this mean they are likely to be more socioeconomically privileged that the general Brazilian population? What does this mean for your results? Please comment.
How does participants household crowning, health status and income compare with the general population?
If the participants are not comparable to the general population, that is fine, but readers need to understand why and how.
Line 258 – convoluted, please edit for clarity.
How cost effective do you think this approach to familial screening is? Please comment.
Author Response
Belo Horizonte, December 14, 2021
Mr. Daniel Iosub
Section Managing Editor, MDPI AG
Pathogens
Dear Mr. Iosub,
Attached, you will find attached a revised copy of the original article: “Evaluation Of Familial Risk of Rheumatic Heart Disease With Systematic Echocardiographic Screening: Data From the PROVAR+ Family Study” that we are submitting for reconsideration in Pathogens.
We have worked on the manuscript to improve general and specific points highlighted by the reviewers. Modifications suggested by the editor and the reviewers were made and the questions were answered and listed below in this letter, and figures to clarify statistical aspects if the analysis were included. The modifications that we made to the text are indicated in each answer.
The positive comments helped us improve the quality of the report. We hope that the revised manuscript is now acceptable for publications.
I would like to assure you that all authors participated actively in this study. All of them have seen and approved the submitted manuscript, which reports unpublished work not under consideration elsewhere. There are no conflicts of interest and the study complies with current ethical considerations.
# Reviewer 1:
Major comments
This is largely a well written manuscript which demonstrates a need for low-cost familial echo screening among first degree relatives of RHD cases in Brazil. The methodology and conclusions appear appropriate.
A: Thank you for revising our manuscript and for the positive comments. Please see our edits throughout the text, according to your recommendations.
The Introduction and Methods are not sufficiently detailed. In places the presentation of Results in convoluted.
A: We have worked on several points of the manuscript, including some details highlighted in red in the Introduction. Please let us know if anything else is needed prior to publications.
In places the wording does not flow well. I have noted a few examples where this occurs below. The manuscript needs a good copy-edit from a naturalized English speaker.
A: A full revision of the manuscript was performed by a native speaker, and several edits were made to improve understanding.
Minor comments
There is a typo in the last author’s degree.
A: We double-checked, and last author’s affiliation is correct: MD (medical doctor) and PhD (Doctor of Philosophy), according to commonly used standards for such degrees.
Abstract:
RHD should be spelled in full at first use in the Abstract (and again in the Introduction).
A: This was a typo, and Rheumatic Heart Disease is now expanded in the Abstract, as suggested (line 30, 21).
If there are 47 controls and 121 patients then matched controls were not enrolled in a 1:5 fashion. Please correct throughout manuscript.
A: We enrolled 226 relatives, and 46 controls, resulting in a 4.91 fashion, compatible with the prespecified aim of 1:5. Thus, no amendments were necessary.
Introduction
Poor English – what is meant by ‘improvement of RHD epidemiology’?
A: In accordance with your suggestion, the sentence was rephrased to: “The epidemiological improvement of RHD over the past decades…”, meaning that the improvement of RHD prevalence, burden and mortality was not equally distributed worldwide: a striking reduction was observed in high-income countries, while a stable or worsening pattern occurred in several low and middle-income counties, where over 80% of children in socioeconomic disadvantage live.
Likewise ‘in lower bounds of the socioeconomic development’ is not worded correctly.
A: The sentence was changed to: “that parallels countries with worse indexes of socioeconomic development”, following your suggestion.
More information about the epidemiology of rheumatic fever and RHD and case management in the area the study is set in is needed.
A: More detailed information about these topics was provided in the Introduction section, lines 62 – 69.
Introduce the University Hospital. Where is it located? What population does it treat? How many admissions/year? Is it a tertiary hospital?
A: More detailed information about the University Hospital (“…a quaternary public institution with 509 beds and 1,500 admissions/month, located in Belo Horizonte, MG, Southeast Brazil”) was provided in the Methods section, lines 104 – 105.
Methods
What us meant by ‘Data analytic methods’? Statistical analyses should be described in the manuscript. The first sentence is odd. Is there a Data Availability section where it could appear instead?
A: This sentence was moved down to a Data Availability Statement section, after the Acknowledgements. This is a commonly used sentence to state that the authors are able to provide raw data for replicating the study’s procedures, upon a reasonable request.
State The PROVAR+ study aims and consider introducing it in the Introduction. Is this the PROVAR+ study or a sub-study?
A: More detailed information about the PROVAR+ study was added accordingly, to the first paragraph of the Methods section (lines 97 – 100). We informed about the first phases of the study, focused on RHD screening in schoolchildren, and then the incorporation of screening into primary care. In the second paragraph, first sentence, we made it clear that this is a sub-study of the PROVAR+ program.
‘study flowchart’ line 78 is an inappropriate description.
A: For better understanding, “flowchart” was replaced by “methodology”.
Who were the non-physicians performing screening? How were they selected and trained?
A: The screening team consisted of one nurse and one physical therapist, previously trained with a combination of online RHD educational modules (http://www.wiredhealthresources.net/EchoProject/index.html) and at least 6 weeks of hands-on training. This information was provided in the second paragraph of the Methods section (lines 113 – 117).
What sort of patient consent was obtained? Eg. Informed written consent following an in-person consultation (and with whom)? I note this is specified below in line 109, suggest deleting above consent section and expanding it here.
A: Consent was obtaining through the signature of an informed written consent form, presented to the patient in the first in-person consultation. This sentence was adjusted and moved to the 2nd paragraph of the Methods section, as suggested.
Who were the ‘experts’? How were they selected and trained?
A: Experts were board-certified cardiologists from the University’s staff, trained in previous phases of the study. This information was added to the Methods, lines 109 – 110, as suggested.
How was socioeconomic matching performed? What factors were matched on? How were these measured?
A: Socioeconomic matching was based on household/socioeconomic environment. We assumed that non-relatives that live in the same house or family compound share similar access to sanitation, healthcare and education, and this represents a similar socioeconomic background for the purposes of the study – to evaluate RHD prevalence. This was made clearer in the sentence: “Socioeconomically-matched controls (spouses, neighbors living in the same family compound) living in the same household for at least 5 years, and thus sharing a similar socioeconomic environment) were also enrolled in a 1:5 fashion”.
Figure 1 should state the no. discharged
A: The figure was changes, and the number discharged was added to the specific box, as suggested. Also, the “controls” box was disconnected from the “relatives” box.
Where was echo screening performed? In participants’ homes? How? Eg. Patients were asked to lie on their bed while screening took place?
A: Screening was performed in the outpatient clinics, during regular appointments, utilizing examination stretchers. This information was added to the Methods section.
Line 106 – capitalize University Hospital.
A: This change was made to the text.
Line 125 ‘at total’ suggest replacing with ‘a total of..’
A: This change was made to the text.
The prevalence of hypertension, diabetes and previous stroke in your study sample is really high. Mention in the Discussion whether this is normal in the regional population. The way you have presented this, it looks like you have grouped the cases in with these prevalence results as well. Please clarify who these results pertain to.
A: Thank you for this remark. We included a 7th paragraph in the Discussion section, discussing characteristics of our sample. Compared to previous publications – cited in this paragraph – the prevalence of hypertension was compatible, especially in the control group. Although the disease may impact valve disease severity and progression (according to the GBD modelling), the rates were not particularly high. We’d like to inform, however, that the rates in the text were incorrect (and different from Table 1), due to a typo, now corrected. In addition, the cases were not grouped based on risk factors. About stroke, we believe that the higher rates in the relatives group was associated with the observed prevalence of valve disease.
‘Among 1st 127 degree relatives, 26 (11.5%) had more than 1 family member with history of RHD and, 128 among index cases, 70.8% reported past prescription of Benzathine Penicillin B (BPG), and 129 only 21.1% are currently under prescription.’ This sentence is convoluted, break it into 2 sentences addressing different things.
A: This change was made to the text, in accordance with your suggestion. The sentence was split in two: the first referring to the relatives, and the second to the index RHD cases.
Be constant with abbreviations – state it in full once in the abstract and main text, then abbreviate consistently eg. mitral valve.
A: Abbreviations were amended throughout the text, as suggested.
Table 1
Age – specify in years
A: This information was added to Table 1.
What does ‘household ‘ refer to? No. people dwelling in home? If you had the number of bedrooms, this information could be combined into a more meaningful assessment of household crowning. Consider using the Canadian National Occupancy Index if data is available.
A: Household refers to number of people sharing the same home (dwelling in home). Unfortunately we don’t have the number of bedrooms, as we did not do this assessment. Thus, the utilization of the Canadian National Occupancy Index was not possible.
Mother education – More information is required to interpret what this line is referring to.
A: This refers to the level of education of the mothers of individuals undergoing echo screening, as a, additional surrogate for socioeconomic status. This was made clearer in the sentence (“Maternal literacy of screened relatives was overall low, with a high proportion of illiterates and/or with incomplete elementary school”) and was added to the footnotes of Table 1.
How was ‘recurrent pharyngitis’ defined? (State in Methods).
A: Recurrent pharyngitis was defined as more than 1 episode occurring in 1 year. Again here, this data was based on self-reports by the patients and/or relatives. This information was added to the subheading of Table 1.
Was household income assessed? How did it compare with the Brazilian median income?
A: Unfortunately, we did not access specifically household income, as this data was prone to inaccuracies. We considered the definition of low-income areas based on the local Human Development Index the vulnerability information from the local boards of health. Thus, from our data it is not possible to make inferences about the comparison with the Brazilian median income.
Table 2.
Suggest using ‘Echocardiographic findings’ in title instead of ‘characteristics’
A: This change was made to the title of Table 2.
I don’t think the total column adds much. Suggest deleting it.
A: As suggested, the total column was excluded from both Tables.
I suggest avoiding using abbreviations in the tables unless it’s really necessary
A: We only kept the most commonly used abbreviations in the tables. Abbreviations related to mitral and aortic valves, coronary artery disease and heart failure were excluded.
Lines 152-158 – there is no need to repeat what is in the table in the text.
A: We left the most essential information in the text, and the results repeated from the Tables are the ones essential for understanding the key characteristics of the included sample. Please let us know if additional edits are necessary.
In the text, spell numbers under 10 in full, Roman numerals can be used for no.s with decimal places and >10. (Line 159)
A: These corrections have been made throughout the text and are highlighted in red.
Line 159 – was this the same patient with mild left ventricular dysfunction and indication of commissurotomy for MV stenosis? Specify. Break into 2 sentences if different patients.
A: The patients are different, and the sentences were broken in two as suggested.
Figure 2 - commas in the caption are unnecessary.
A: The commas were removed from the caption of Figure 2.
Discussion
Line 172 – rewrite, ‘in its majority,’ this does not make sense.
A: For clarification, the sentence was changed to “RHD phenotypes at diagnosis varied from predominantly subclinical valvular disease, to advanced…”.
Line 174 reference all these studies
A: The 2 studies that evaluated family screening for RHD were referenced in this sentence.
Reference 20 is formatted oddly and you seem to be missing reference 19.
A: We have double-checked the references, and they are now correct, according to the MDPI style in the EndNote software.
Reference all of the studies you have referred to.
A: We have referenced all studies we referred to, throughout the manuscript.
Line 193 – re-write. What is meant by ‘its strongest predictors’?
A: It means the strongest predictors of “clinical disease” or clinical RHD. This was made clearer in the text, after rephrasing this sentence.
Reference 22 – in what population?
A: Where reported, the population of the studies included in this meta-analysis was mostly from North America and Ireland. We included this information in the aforementioned sentence.
‘Contrasting with this sample of clinically diagnosed disease..’ rephrase.
A: Following your suggestion, the sentence was rephrased to: “Contrasting with this sample of individuals with advanced disease and clinical manifestations…”.
Line 216 – ‘…inclusion of non-relatives sharing the same environment for a considerable time period…’ please define the min. time in shared environment for controls to be eligible in the Methods. Do you mean shared household not environment?
A: The time period (at least five years) considered for shared household was stated in the Methods section and also in this sentence, in Discussion. We meant shared household, meaning that a similar social environment was shared by the non-relatives.
Line 245 – at first? What changed? Rewrite?
A: The sentence was adjusted, as suggested. At first was changed to “first”, meaning the number 1 limitation stated in this section.
Your cases were selected from people who have access to a tertiary specialized outpatient clinic. Does this mean they are likely to be more socioeconomically privileged that the general Brazilian population? What does this mean for your results? Please comment.
How does participants household crowning, health status and income compare with the general population? If the participants are not comparable to the general population, that is fine, but readers need to understand why and how.
A: About these questions, we included an entire paragraph in the Discussion session (7th paragraph), discussing the access to tertiary care of our population, which may reflect a more privileged subgroup of the Brazilian population. However, we additionally included – as suggested – a comparison of the median household in our sample with the general population (Brazilian average). The rates in our sample were considerably higher, as an additional evidence of low socioeconomic status and overcrowding. Please let us know if additional points are necessary to this discussion.
Line 258 – convoluted, please edit for clarity.
A: The sentence was edited and a full revision of the manuscript by a native English speaker was performed. Please let us know if any additional edits are necessary.
How cost effective do you think this approach to familial screening is? Please comment.
A: Unfortunately to date there is no information regarding cost-effectiveness of family screening for RHD. Screening for RHD in schools from underserved areas seem to be cost-effective, with publications from Brazil, Oceania and Africa, but data is still insufficient. This limitation was added to the Discussion and Limitations sections.
Reviewer 2 Report
Overall quality of manuscript is low-average. Nevertheless, I think some MAJOR REVISIONS should be necessary before considering acceptance by Editorial Team.
Among main POINTs of WEAKNESSES we can insert:
- English level. To be checked (i.e. lines 27-29 and 46-48: sentences might be globally revised).
- Quality of tables is poor. PLEASE TRY to IMPROVE them!
- About METHODs, what about inter- and intra- observer (for echo) variability? PLEASE EXPLAIN.
- Moreover, I'd like to suggest some readings in order to enlarge INTRODUCTION and partly DISCUSSION/CONSLUSIONS:
- Front Med (Lausanne). 2018 Feb 14;5:26. doi: 10.3389/fmed.2018.00026. eCollection 2018.
- Atherosclerosis. 2015 Jul;241(1):259-63. doi: 10.1016/j.atherosclerosis.2015.03.044. Epub 2015 Apr 3.
- Autoimmun Rev. 2016 Jul;15(7):756-69. doi: 10.1016/j.autrev.2016.03.014. Epub 2016 Mar 12.
- And finally, please check all references becasuse they are not always in accordance with author's rules.
Best regards,
Author Response
Mr. Daniel Iosub
Section Managing Editor, MDPI AG
Pathogens
Dear Mr. Iosub,
Attached, you will find attached a revised copy of the original article: “Evaluation Of Familial Risk of Rheumatic Heart Disease With Systematic Echocardiographic Screening: Data From the PROVAR+ Family Study” that we are submitting for reconsideration in Pathogens.
We have worked on the manuscript to improve general and specific points highlighted by the reviewers. Modifications suggested by the editor and the reviewers were made and the questions were answered and listed below in this letter, and figures to clarify statistical aspects if the analysis were included. The modifications that we made to the text are indicated in each answer.
The positive comments helped us improve the quality of the report. We hope that the revised manuscript is now acceptable for publications.
I would like to assure you that all authors participated actively in this study. All of them have seen and approved the submitted manuscript, which reports unpublished work not under consideration elsewhere. There are no conflicts of interest and the study complies with current ethical considerations.
# Reviewer 2:
Among main POINTs of WEAKNESSES we can insert:
A: Thank you for revising our manuscript, and for the positive comments about it. We have worked to edit the text/tables to address the points highlighted by the reviewers, as detailed below.
English level. To be checked (i.e. lines 27-29 and 46-48: sentences might be globally revised).
A: The manuscript was fully revised by a native English speaker. In addition, this specific session (Introduction) was fully revised grammatically, and some new data requested by reviewer 1 were included.
Quality of tables is poor. PLEASE TRY to IMPROVE them!
A: As suggested, we improved the quality of the tables by: a) removing the “Total” column (with the combined sample relatives + controls), as it did not add to the understanding of the manuscript. Information about some variables (recurrent pharyngitis, mother education) was added to the subheadings for clarifications. Finally, the number of abbreviations in the tables was considerably reduced.
About METHODs, what about inter- and intra- observer (for echo) variability? PLEASE EXPLAIN.
A: We re-analyzed our data, and included the Kappa test to evaluate inter-observer variability for the interpretation of screening echos. The agreement for the diagnosis of any RHD was high, resulting a Kappa = 0.89. This was included in the Methods and Results section. The agreement between screening / standard echo had been previously reported in the manuscript (82%).
Moreover, I'd like to suggest some readings in order to enlarge INTRODUCTION and partly DISCUSSION/CONCLUSIONS:
- Front Med (Lausanne). 2018 Feb 14;5:26. doi: 10.3389/fmed.2018.00026. eCollection 2018.
- Atherosclerosis. 2015 Jul;241(1):259-63. doi: 10.1016/j.atherosclerosis.2015.03.044. Epub 2015 Apr 3.
- Autoimmun Rev. 2016 Jul;15(7):756-69. doi: 10.1016/j.autrev.2016.03.014. Epub 2016 Mar 12.
A: Thank you for these suggestions. We have incorporated in the Discussion section (2nd paragraph), especially the first manuscript about imaging modalities in inflammatory diseases, including those of rheumatic etiology. The other references refer to early atherosclerosis in systemic inflammatory diseases, which is not the specific case of Rheumatic Heart Disease, the topic of our investigation. Please let us know if you have additional suggestions.
And finally, please check all references because they are not always in accordance with author's rules.
A: We have double-checked all the references, utilizing the EndNote software and manually, to confirm that they are in accordance with the author’s rules.
Corresponding author:
Bruno Ramos Nascimento, MD, MSc, Ph.D, FACC, FESC
Associate Professor of Medicine
Hospital das Clínicas da Universidade Federal de Minas Gerais
American Heart Association Professional Membership ID: 210403212
Rua Muzambinho 710, apt. 802, Serra
Belo Horizonte, Minas Gerais, Brasil, CEP 30.210-530
Tel.: +55 31 3409 9437; Fax: +55 31 32847298.
E-mail: ramosnas@gmail.com
Twitter: @ramosnas
Reviewer 3 Report
Reviewing the manuscript entitled, “Evaluation Of Familial Risk of Rheumatic Heart Disease With Systematic Echocardiographic Screening: Data From the 3 PROVAR+ Family Study” Franco J et al., this is focusing on relevance between RHD and familial risk particular 1st degree relatives. RHD is a disease that has problems to be solved such as establishment of a screening method for early detection as a disease to be solved in the poor. Therefore, although the number of n is small, research from such a new perspective is important. However, A comparison between the two groups alone does not deserve medical implications. The authors need to respond to following concerns for acceptable quality.
From line 118 to 119, although you mentioned “Demographic, clinical and echocardiographic variables 118 were compared between groups”, what does it mean? Have you verified the results using multivariate analysis? There is no such description in the paragraph of statistical analysis. No matter how significant the difference between the two variables is statistically analyzed, it cannot be a medical verification. The authors should perform multivariate analysis with the factors that influence the outcome as covariates. And then you should modify section of the results and the discussion including limitations.
The authors should modify Figure 1. What do three arrows from Index (N=121): advanced RHD valve disease mean? Why is there no arrow to parents?
Author Response
Belo Horizonte, December 14, 2021
Mr. Daniel Iosub
Section Managing Editor, MDPI AG
Pathogens
Dear Mr. Iosub,
Attached, you will find attached a revised copy of the original article: “Evaluation Of Familial Risk of Rheumatic Heart Disease With Systematic Echocardiographic Screening: Data From the PROVAR+ Family Study” that we are submitting for reconsideration in Pathogens.
We have worked on the manuscript to improve general and specific points highlighted by the reviewers. Modifications suggested by the editor and the reviewers were made and the questions were answered and listed below in this letter, and figures to clarify statistical aspects if the analysis were included. The modifications that we made to the text are indicated in each answer.
The positive comments helped us improve the quality of the report. We hope that the revised manuscript is now acceptable for publications.
I would like to assure you that all authors participated actively in this study. All of them have seen and approved the submitted manuscript, which reports unpublished work not under consideration elsewhere. There are no conflicts of interest and the study complies with current ethical considerations.
# Reviewer 3:
Reviewing the manuscript entitled, “Evaluation Of Familial Risk of Rheumatic Heart Disease With Systematic Echocardiographic Screening: Data From the 3 PROVAR+ Family Study” Franco J et al., this is focusing on relevance between RHD and familial risk particular 1st degree relatives. RHD is a disease that has problems to be solved such as establishment of a screening method for early detection as a disease to be solved in the poor. Therefore, although the number of n is small, research from such a new perspective is important. However, A comparison between the two groups alone does not deserve medical implications. The authors need to respond to following concerns for acceptable quality.
A: Thank you for the positive comments about our study, and for reviewing our manuscript. We have edited the manuscript, addressing the points highlighted by the 3 reviewers accordingly. We agree that sample size was a limitation, precluding a more detailed statistical approach, but the investigation shows a new perspective for targeted echocardiographic screening for RHD, especially in resource-limited settings as Latin America.
From line 118 to 119, although you mentioned “Demographic, clinical and echocardiographic variables 118 were compared between groups”, what does it mean? Have you verified the results using multivariate analysis? There is no such description in the paragraph of statistical analysis. No matter how significant the difference between the two variables is statistically analyzed, it cannot be a medical verification. The authors should perform multivariate analysis with the factors that influence the outcome as covariates. And then you should modify section of the results and the discussion including limitations.
A: We agree that the absence of multivariable analysis precludes more robust extrapolations from our data, and direct clinical application of our findings. However, multivariable analyses of RHD predictors was not possible due to a) the relatively small sample size, due to the difficulty to include relatives of patients with RHD, who are usually not enrolled in clinical follow-up in tertiary centers; b) the small number of events (presence of screen-detected RHD). Thus, a multivariate logistic model was not possible. But, we totally agree that this is a limitation, and now it’s clearly stated in the Limitations section (first limitation). Please let us know if any additional edits are necessary.
The authors should modify Figure 1. What do three arrows from Index (N=121): advanced RHD valve disease mean? Why is there no arrow to parents?
A: The Figure 1 was edited, as suggested: the first box was changed to “Index cases (N=121…), meaning that these are the index patients with advanced RHD, whose relatives and related non-relatives were invited for screening. The arrow was inserted in the Parents box, and the Controls box was disconnected from the Index Cases.
Corresponding author:
Bruno Ramos Nascimento, MD, MSc, Ph.D, FACC, FESC
Associate Professor of Medicine
Hospital das Clínicas da Universidade Federal de Minas Gerais
American Heart Association Professional Membership ID: 210403212
Rua Muzambinho 710, apt. 802, Serra
Belo Horizonte, Minas Gerais, Brasil, CEP 30.210-530
Tel.: +55 31 3409 9437; Fax: +55 31 32847298.
E-mail: ramosnas@gmail.com
Twitter: @ramosnas
Round 2
Reviewer 2 Report
I have much appreciated authors' job on original manuscript.
Author Response
Belo Horizonte, December 27, 2021
Mr. Daniel Iosub
Section Managing Editor, MDPI AG
Pathogens
Dear Mr. Iosub,
Attached, you will find attached the second revised copy of the original article: “Investigation Of Familial Risk of Rheumatic Heart Disease With Systematic Echocardiographic Screening: Data From the PROVAR+ Family Study” that we are submitting for reconsideration in Pathogens.
We have worked on the manuscript to improve general and specific points highlighted by the reviewers. Modifications suggested by the editor and the reviewers were made and the questions were answered and listed below in this letter, and figures to clarify statistical aspects if the analysis were included. The modifications that we made to the text are indicated in each answer.
The positive comments helped us improve the quality of the report. We hope that the revised manuscript is now acceptable for publications.
I would like to assure you that all authors participated actively in this study. All of them have seen and approved the submitted manuscript, which reports unpublished work not under consideration elsewhere. There are no conflicts of interest and the study complies with current ethical considerations.
I look forward to hearing from you.
Sincerely yours,
Bruno R Nascimento
# Reviewer 2:
I have much appreciated authors' job on original manuscript.
Suggestions for Authors: none reported.
A: Thank you for the positive review, and for the suggestions that really helped us improve the quality of the paper. Please let us know if any additional edits are necessary.
Corresponding author:
Bruno Ramos Nascimento, MD, MSc, Ph.D, FACC, FESC
Associate Professor of Medicine
Hospital das Clínicas da Universidade Federal de Minas Gerais
American Heart Association Professional Membership ID: 210403212
Rua Muzambinho 710, apt. 802, Serra
Belo Horizonte, Minas Gerais, Brasil, CEP 30.210-530
Tel.: +55 31 3409 9437; Fax: +55 31 32847298.
E-mail: ramosnas@gmail.com
Twitter: @ramosnas
Reviewer 3 Report
Reviewing the manuscript entitled, “Evaluation Of Familial Risk of Rheumatic Heart Disease With Systematic Echocardiographic Screening: Data From the 3 PROVAR+ Family Study” Franco J et al., this is focusing on relevance between RHD and familial risk particular 1st degree relatives. Again, there is no medical implication unless the effects of independent variables are eliminated by performing multivariate analysis. On the other hand, I understand that the number of samples is not enough to perform multivariate analysis sufficiently. Therefore, the authors need to modify the title. Your tile includes “Evaluation”. Usually, in case of “Evaluation” in the clinical study, it means medical implication from the results. So your title is no adequate for this manuscript.
Author Response
Belo Horizonte, December 27, 2021
Mr. Daniel Iosub
Section Managing Editor, MDPI AG
Pathogens
Dear Mr. Iosub,
Attached, you will find attached the second revised copy of the original article: “Investigation Of Familial Risk of Rheumatic Heart Disease With Systematic Echocardiographic Screening: Data From the PROVAR+ Family Study” that we are submitting for reconsideration in Pathogens.
We have worked on the manuscript to improve general and specific points highlighted by the reviewers. Modifications suggested by the editor and the reviewers were made and the questions were answered and listed below in this letter, and figures to clarify statistical aspects if the analysis were included. The modifications that we made to the text are indicated in each answer.
The positive comments helped us improve the quality of the report. We hope that the revised manuscript is now acceptable for publications.
I would like to assure you that all authors participated actively in this study. All of them have seen and approved the submitted manuscript, which reports unpublished work not under consideration elsewhere. There are no conflicts of interest and the study complies with current ethical considerations.
I look forward to hearing from you.
Sincerely yours,
Bruno R Nascimento
# Reviewer 3:
Reviewing the manuscript entitled, “Evaluation Of Familial Risk of Rheumatic Heart Disease With Systematic Echocardiographic Screening: Data From the 3 PROVAR+ Family Study” Franco J et al., this is focusing on relevance between RHD and familial risk particular 1st degree relatives. Again, there is no medical implication unless the effects of independent variables are eliminated by performing multivariate analysis. On the other hand, I understand that the number of samples is not enough to perform multivariate analysis sufficiently. Therefore, the authors need to modify the title. Your tile includes “Evaluation”. Usually, in case of “Evaluation” in the clinical study, it means medical implication from the results. So your title is no adequate for this manuscript.
A: Thank you for reviewing the revised version of our manuscript. We agree that “Evaluation” may indicate some clinical implication of the study findings, and such inference is limited by the absence of multivariable analysis of predictors of scree-detected RHD. As you pointed out, and as stated in the Limitations section, this limitation was due to the relatively limited sample size. Thus, following your recommendation, we changed “Evaluation” for “Investigation”, giving a more scientific and less clinical assessment of family risk of RHD.
In addition, we performed additional checking of the English language.
Corresponding author:
Bruno Ramos Nascimento, MD, MSc, Ph.D, FACC, FESC
Associate Professor of Medicine
Hospital das Clínicas da Universidade Federal de Minas Gerais
American Heart Association Professional Membership ID: 210403212
Rua Muzambinho 710, apt. 802, Serra
Belo Horizonte, Minas Gerais, Brasil, CEP 30.210-530
Tel.: +55 31 3409 9437; Fax: +55 31 32847298.
E-mail: ramosnas@gmail.com